# Towards Deep Cellular Phenotyping in Placental Histology

**Michael Ferlaino**[*]
Big Data Institute
University of Oxford
michael.ferlaino@bdi.ox.ac.uk

**Craig A. Glastonbury**[*]
Big Data Institute
University of Oxford
craig@well.ox.ac.uk

**Carolina Motta-Mejia**
Nuffield Department of Women's
& Reproductive Health
University of Oxford

**Manu Vatish**
Nuffield Department of Women's
& Reproductive Health
University of Oxford

**Ingrid Granne**
Nuffield Department of Women's
& Reproductive Health
University of Oxford

**Stephen Kennedy**
Nuffield Department of Women's
& Reproductive Health
University of Oxford

**Cecilia M. Lindgren** [†‡§]
Big Data Institute
University of Oxford

**Christoffer Nellåker**
Big Data Institute
Nuffield Department of Women's
& Reproductive Health
University of Oxford

## Abstract

The placenta is a complex organ, playing multiple roles during fetal development. Very little is known about the association between placental morphological abnormalities and fetal physiology. In this work, we present an open sourced, computationally tractable deep learning pipeline to analyse placenta histology at the level of the cell. By utilising two deep convolutional neural network architectures and transfer learning, we can robustly localise and classify placental cells within five classes with an accuracy of 89%. Furthermore, we learn deep embeddings encoding phenotypic knowledge that is capable of both stratifying five distinct cell populations and learn intraclass phenotypic variance. We envisage that the automation of this pipeline to population scale studies of placenta histology has the potential to improve our understanding of basic cellular placental biology and its variations, particularly its role in predicting adverse birth outcomes.

## 1   Introduction

Despite its essential functions, placenta biology remains poorly understood [1]. For instance, we lack sufficient understanding of the molecular mechanisms linking cell composition and tissue

---

[*] **These authors contributed equally to this work.**
[†]Li Ka Shing Centre for Health Information and Discovery, University of Oxford
[‡]Wellcome Trust Centre for Human Genetics, University of Oxford
[§]Program in Medical and Population Genetics, Broad Institute, Cambridge, Massachusetts, USA

1st Conference on Medical Imaging with Deep Learning (MIDL 2018), Amsterdam, The Netherlands.

morphology to fetal growth and development. Previous research suggests there exists a direct link between placental disease and adverse effects on fetal physiology, some of which have been found to be associated with histopathological evidence of abnormality [2, 3, 4]. This supports the idea that histological phenotyping of placental tissues could fill gaps in our understanding of placental health and birth outcomes.

In this work we start addressing some of these fundamental issues by developing a deep learning pipeline to automate the analysis of cell phenotyping in placenta histology. By exploiting convolutional neural networks (CNNs) and transfer learning [5], we implement a two part workflow that can accurately localise and objectively classify cell populations in placenta imaging. Furthermore, we learn deep embeddings capable of capturing intercellular morphological variance as well as intraclass phenotypic heterogeneity. Our approach analyses placenta imaging by solving two distinct computer vision tasks. For each test image, we firstly detect nuclei by means of the nuclei localisation module; the classification module then phenotypes and segregates cells within five populations present in the placenta at term.

## 2 Related Work

Histology is routinely performed as a diagnostic test, intraoperative guide and tool for staging disease severity [6]. However, in the era of big data, little automation is in place to enable histological analysis to scale to thousands of samples, making histology a manual and time intensive task. Over the last two decades, extensive research has focused on the automatic detection of pathological lesions in histology samples. Classic computer vision methods, such as difference of Guassians and Gabor filters for nuclei detection [7], Watershed and Otsu thresholding for segmentation [8], or generic feature extractors such as CellProfiler [9], have been superseded by deep learning approaches [10, 11]. This methodological shift from handcrafted features to learned representations has led to the current state of the art performance in multiple cognitive tasks, e.g. nuclei segmentation [12], region of interest detection [13], and pathological lesion classification [14].

Whilst nuclei detection is well documented, very few studies have focused on the classification of specific, well annotated cell types curated from histology imaging data [15]. Furthermore, even though cellular and tissue context are important, classification has been performed at the level of the nuclei rather than cells [16]. This lack of cellular annotations and classification approaches is primarily due to the need for expert annotation, an expensive and time consuming process. Several studies have tried to avoid the need for annotations by learning representations with unsupervised approaches [17].

In the present study, we provide a data set comprised of thousands of high confidence and manually curated placenta cells spanning five classes. These examples, obtained from healthy human samples, were used to demonstrate how fully supervised representations encode biological and morphological knowledge at the cellular level.

## 3 Methods

### 3.1 Data Collection

We curated ten samples of formalin embedded placental sections, collected from healthy women [18] who gave birth at term to healthy, appropriately grown babies based on the INTERGROWTH-21st newborn size standards [19]. Each tissue section was embedded in paraffin and microtome sliced to a thickness of $5\,\mu\mathrm{m}$, before being stained with Haematoxylin and Eosin (H&E) [20]. The samples were digitised into whole slide images (WSIs) using a Zeiss Axio Scan Z1 system (manufacturer) at $40\times$ magnification. Each individual WSI was then split into smaller subimages, termed tiles, of $1200 \times 1600$ pixels. Lastly, as a preprocessing step, all tiles containing image artefacts (*e.g.* tissue folds and blurred regions) were filtered out. The project was approved by the Oxfordshire Research Ethics Committee "C" (reference: 08/H0606/139).

### 3.1.1 Nuclei Localisation Data

We used the VGG image annotator [21] to curate 91 tiles (subimages of $1\,200 \times 1\,600$ pixels). Across all images, discernible nuclei were annotated with bounding boxes, for a total of $13\,179$ examples.

All data were then split into training (11 184 nuclei), validation (876 nuclei), and test (1 119) sets. Crucially, training, validation, and test tiles were sampled from different individuals (WSIs).

### 3.1.2  Cell Classification Data

Cells present in term placentas belong to one of five different types: cytotrophoblast (CYT), fibroblast (FIB), Hofbauer (HOF), syncytiotrophoblast (SYN), and vascular (VAS). For the classification problem, data sets were annotated by randomly sampling tiles across individuals. Data were manually curated with the help of two senior obstetricians (MV, IG) who annotated and provided training for discriminating cell types within placenta histology images. After nuclei localisation, each tile was presented to one annotator in order to assign (ground truth) cell type labels to detected nuclei. To reduce the number of false positives, labels were assigned only for nuclei that could be discriminated, with high confidence, by annotators. Data were curated using the VGG image annotator.

We annotated 11 666 tiles for a total of 9 529 ground truth labels. For each cell annotated, a $200 \times 200$ pixels patch, centred at the location of the nucleus, was generated. Such patches, along with the corresponding cell type labels, comprised the data sets used for training CNNs.

Data were randomly split into training, validation, and test sets. We annotated balanced validation and test sets, both comprised of 1 000 images (200 examples per class). All remaining images were used to generate a training sample of 7 529 instances (1 359 CYTs, 2 577 FIBs, 478 HOFs, 1 576 SYNs, 1 539 VASs). Figure S1 visualises the morphological heterogeneity across cell types.

### 3.2  Deep Learning Pipeline

Our framework can analyse placenta histology images by exploiting deep learning modules to solve two visual tasks. For each test tile, we first localise and store the coordinates of all nuclei identified in the image. Then, for each coordinate, we create a patch of $200 \times 200$ pixels centered around each nucleus. This image is then provided as input to the classification module in order to predict the type of cell present in the patch (Figure 1).

Due to the relatively small size of our training sets, and the complexity of the visual tasks we are trying to solve, our approach relies heavily on transfer learning and fine tuning. These approaches utilise visual representations encoded by CNNs trained on massive data sets, like ImageNet [23] and COCO [24], to solve multiple vision recognition problems, beyond the original learning task [5].

### 3.2.1  Nuclei Detection

For the nuclei detection task, we experimented with two different approaches. We trained a fully convolutional neural network (FCNN), which learns nuclei density maps, and a recent localisation framework (RetinaNet) published by Facebook [22], which regresses bounding boxes for each target nucleus.

The fully convolutional neural network we developed, termed FCNN-Unet, was inspired by the FCNN-A architecture introduced in [25], and is characterised by the addition of skip connections, dropout, and batch normalisation.

Furthermore, we implemented RetinaNet by using a ResNet50 backbone (pretrained on COCO). The major innovation introduced by RetinaNet is the use of a focal training loss. Focal losses generalise the commonly used cross entropy by reducing penalties for well classified examples, thereby allowing the network to focus on hard to predict instances [22].

For each nucleus, RetinaNet outputs bounding box locations, along with posterior probabilities quantifying the network's confidence in the box containing a nucleus. We set an upper bound of 500 for the maximum number of bounding boxes predicted per image. We fine tuned the bounding box regressor with a small learning rate ($10^{-4}$) using Adam, an adaptive optimiser [26]. The learning rate was reduced by $10^{-1}$ whenever there was no improvement in validation accuracy for more than four epochs. Furthermore, we implemented horizontal/vertical flips, scaling and stain transformations.

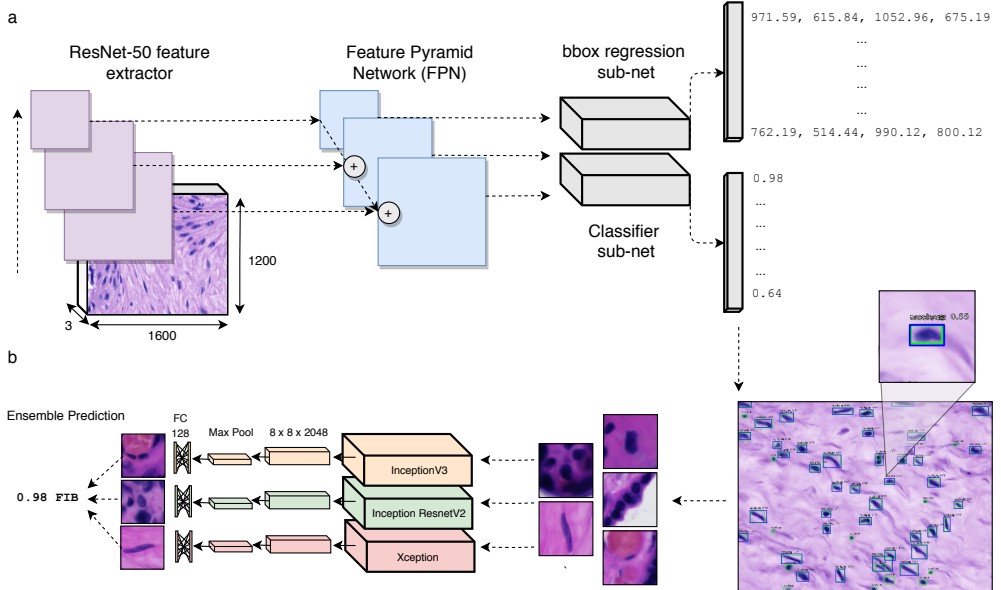

Figure 1: **Deep learning pipeline.** Our framework consists of two deep learning modules. (a) **Nuclei locator.** We use RetinaNet [22] with a ResNet-50 backbone as a feature extractor. Learned representations are fed to subnetworks to perform classification (background vs nucleus) and bounding box regression (nuclei locations). (b) **Cell classifier.** Using the nuclei locations predicted by RetinaNet we obtain cell patches ($200 \times 200$ pixels) which we feed into three independent CNNs. We use an ensemble system to segregate placental cells within five populations by taking the maximum posterior probability from any one of our three base classifiers.

Stain normalisation is a popular data augmentation approach that attempts to minimise H&E stain varability across different digital scanners and laboratory protocols [27]. Most methods involve subjective selection of "reference" images and the use of matrix decomposition approaches such as non-negative matrix factorisation (NMF) [28]. This makes the scaling of such approaches infeasible, especially for whole slide image analyses.

To mitigate the majority of problems caused by high stain variance, we implemented "stain transformations". We converted RGB images into the HED colour scheme, by means of colour deconvolution [29]. Once in the HED space, we randomly scaled the image by a factor uniformly sampled from the interval $(0.95, 1.05)$. This scaling has the effect of simulating either under or over staining of the tissue.

### 3.2.2 Multicell Classification using Transfer Learning

We used InceptionV3 [30, 31] as our base learner, a convolutional neural network architecture developed by Google. As the original task used for training InceptionV3 is highly dissimilar from our target task, we experimented beyond simply using CNNs as feature extractors in order to fine tune (*i.e.* retrain) InceptionV3's layers on our data set.

In order to fine tune InceptionV3, the (original) output layer was removed and InceptionV3's output volume was spatially compressed, by means of max pooling, before being fed to the hidden layer of 128 ReLU neurons. Lastly, the (new) classification layer comprised 5 softmax neurons emitting posterior probabilities of class membership for each cell type (CYT, FIB, HOF, SYN, and VAS).

# 4 Results

## 4.1 Nuclei Localisation using RetinaNet

For solving the nuclei detection task, we implemented both RetinaNet and FCNN-Unet. Both were able to predict nuclei location, but FCNN/FCNN-Unet were unable to learn non-spherical nuclei and erroneously over predicted more than one nucleus for elongated cells (Figure S2 and section 6 for Jupyter notebooks). Further work could explore the impact of learning Gaussians with unequal $\sigma$'s to account for non-spherical nuclei [32].

During RetinaNet training, we implemented stain transformations to account for differences in H&E staining across samples (section 3.2.1). To test the effect of our stain transformation approach, RetinaNet was trained with "standard" data augmentation, as well as with data augmentation supplemented by stain transformations (see section 6). The results of our experimentation are outlined in Table 1. By adding stain transformations to the standard data augmentation protocol, we see a minor improvement in performance across multiple posterior cut offs, especially boosting validation mAP at the default threshold value (0.50). The generalisation performance of the best model was estimated by deploying the network on ten held out test images, achieving an mAP of 0.66. We attribute the discrepancy between validation and test mAPs to be driven by differences in stain concentration across individuals, given that training, validation and test folds were generated from unique individual samples (section 3.1.1), and we did not see any overfitting whilst training with early stopping (Figure S3). Sample stain differences have been recently interpreted as a domain adaptation problem, where the domain shift is in the difference in staining and/or experimental variability between histology slides [33]. Lastly, Figure 2 displays examples of predictions returned by RetinaNet.

| Nucleus Posterior | mAP (A) | mAP (A + S) | Test mAP (A + S) |
|---|---|---|---|
| 0.05 | 0.80 | 0.80 | - |
| 0.25 | 0.78 | 0.79 | - |
| 0.35 | 0.76 | 0.77 | - |
| 0.50 | 0.69 | 0.72 | 0.66 |

Table 1: **RetinaNet performance**. Validation mAP across posterior probability thresholds, with standard data augmentation (A), and with data augmentation supplemented with stain transformations (A + S). We also report RetinaNet's test mAP at the chosen posterior probability cut off.

## 4.2 Deep Learning can Accurately Classify Placental Cells

Our training set is imbalanced, with the majority class (FIB) being more than five times the size of the minority cell type (HOF). Consequently, in order not to learn a biased hypothesis, we tested several methods for generating a balanced training sample from curated data. We experimented with down sampling all classes to the size of the minority class, as well as bootstrapping all cell types to the size of the majority class. We also trained our model by penalising misclassification based on the size of classes, thereby trying to force the network to focus on under represented cell types. Throughout this work we balanced training sets by bootstrapping, as this over sampling approach achieved the highest validation accuracy (Table S1).

We fine tuned *all* layers of our CNN by training for 20 epochs with stochastic gradient descent (batches of 85 images), and using a categorical cross entropy loss. We used dropout (with a rate of $50\%$) before each fully connected layer and deployed an aggressive data augmentation protocol (at run time) by implementing random rotations, horizontal/vertical flips, shear and stain transformations (also see section 6).

Model selection was implemented by saving the network achieving the highest validation accuracy at the end of training. Lastly, generalisation performance was estimated by deploying the CNN on the independent test set. The results of our performance assessment are comprehensively displayed in Figure 3 (confusion matrix) and Table 2 (classification report).

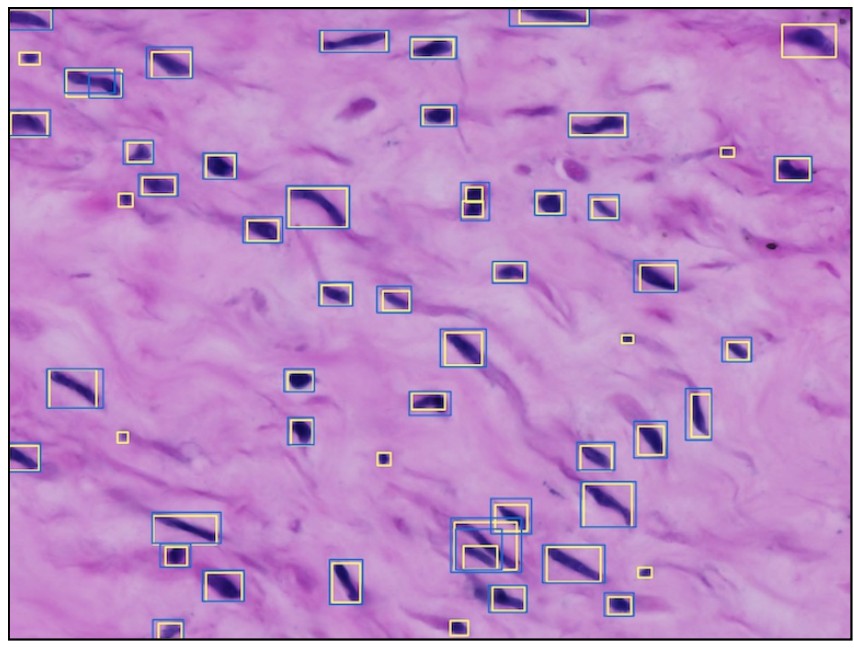

Figure 2: **RetinaNet predictions.** A representative test image, displaying predicted bounding boxes (blue) versus ground truth annotations (yellow).

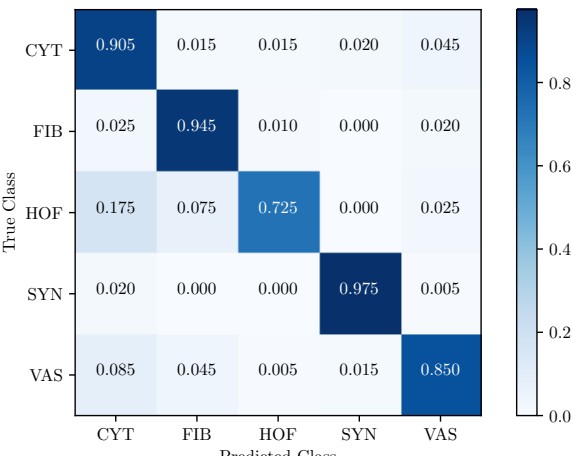

Figure 3: **InceptionV3's confusion matrix.** Visualisation of error rates, on the independent test set, across cell types.

We achieved an overall accuracy of $88\%$. Most cell types were confidently discriminated, with the lowest class accuracy recorded for Hofbauer cells (minority class). HOFs are extremely challenging to annotate in term placentas, as the size of the Hofbauer population peaks early in pregnancy [34]. Despite achieving extremely high precision ($96\%$), our cell classifier could only achieve $73\%$ HOF accuracy because of the high ratio of false negatives (Table 2), as $18\%$ of test HOFs were confused and misclassified as CYTs (Figure 3).

### 4.2.1 Boosting Performance with Ensemble Learning

We developed an ensemble system using InceptionV3, InceptionResNetV2 [35], and Xception [36] as base classifiers. All ensemble members were trained as described in section 4.2 (also see section 6). First of all, somewhat akin to bagging, base learners were trained on different sets generated

| Cell Type | Precision | Recall | *F* Measure |
| --- | --- | --- | --- |
| CYT | 0.748 | 0.905 | 0.819 |
| FIB | 0.875 | 0.945 | 0.909 |
| HOF | 0.960 | 0.725 | 0.826 |
| SYN | 0.965 | 0.975 | 0.970 |
| VAS | 0.899 | 0.850 | 0.874 |
| Average | 0.890 | 0.880 | 0.880 |

Table 2: **Classification report of InceptionV3.** Performance assessment, on test data, estimated by means of precision, recall, and their harmonic mean ($F$ measure). The report also records averaged statistics, across classes, weighted by their support.

by bootstrapping our curated data. In addition, more variation was injected by using different convolutional neural network architectures.

In Table 3, we summarise the results of our experimentation by reporting accuracy scores achieved by each base learner as well as the ensemble system, which selects the cell type predicted by the most confident base learner.

| Model | Validation Accuracy | Test Accuracy |
| --- | --- | --- |
| InceptionV3 | 90% | 88% |
| InceptionResNetV2 | 91% | 87% |
| Xception | 91% | 87% |
| Ensemble (Max) | 91% | 89% |

Table 3: **Performance metrics across models used to develop our cell classifier.** Validation and test accuracies are reported for each of the base cell type classifiers as well as the ensemble model.

Exploiting ensemble learning we were able to improve performance of each base classifier, achieving an overall accuracy of $89\%$. Misclassification error types are comprehensively visualised in Figure S4. Compared to InceptionV3's results (Figure 3), our ensemble system was capable of boosting accuracies across most cell types, crucially reducing the error rate for the minority (HOF) class.

### 4.3   Deploying the Pipeline on Test Tiles

We validated our framework by deploying the pipeline to detect nuclei and predict cell types across whole test images (not used during previous training or validations). This procedure, whose output is displayed in Figure 4, generates visualisations capable of capturing local morphology and cell distributions (additional tiles with superimposed predictions can be found in Figure S5).

It has been shown that heterogenous aggregates of HOFs can be informative descriptors for diagnosing placentas affected by preeclampsia [37]. Therefore, our pipeline deployment across whole tiles, has the potential of being a helpful clinical tool aiding obstetricians in, for instance, estimating local cell population sizes and discovering anomalous cellular aggregates.

### 4.4   Deep Embeddings Capture Intraclass Phenotypic Variance

Our analyses have demonstrated that the representation learned by our ensemble system encodes biological knowledge capable of successfully discriminating cell populations. In this section, we further investigate whether our deep embedding is capable of uncovering phenotypic heterogeneity within cell populations.

To perform such analysis, we exploited dimensionality reduction (DR) techniques to visualise our 384 dimensional embedding on a 2D plane. The vast majority of DR approaches (*e.g.* PCA) implement

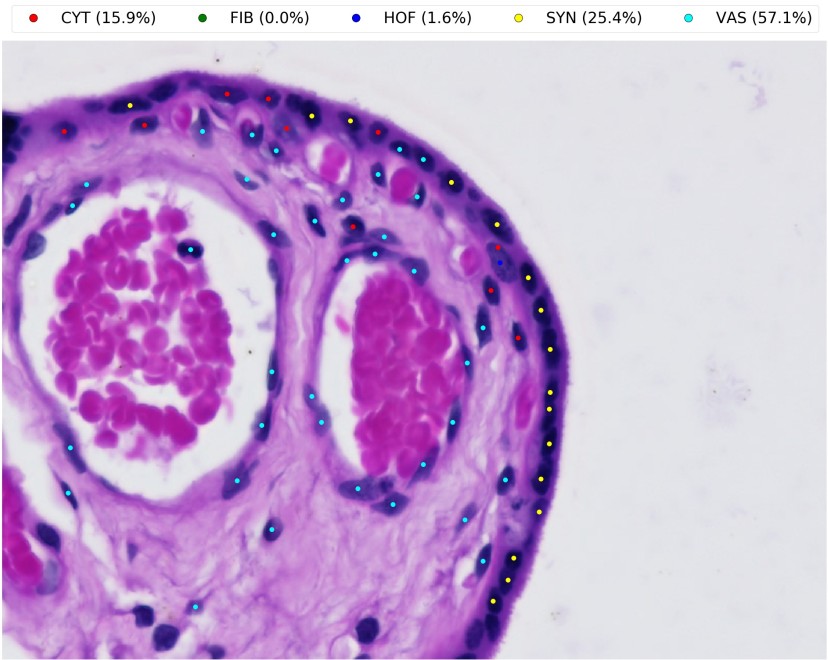

Figure 4: **Pipeline deployment on a test tile.** By deploying our approach for prediction on whole images, we are able to visualise local cellular distributions and population sizes. For instance, the image contains two blood vessels and our pipeline estimated the majority of cells (57.1%) are of vascular type.

linear algorithms and, consequently, can only capture global characteristics of the original feature space [38]. Accordingly, linear methods are insensitive to phenotypic variance expected in placental cell populations. For this reason, we used $t$SNE [38], a nonlinear DR technique capable of preserving global as well as local patterns in our deep representation. This nonlinear method has already been applied to (high dimensional) biological imaging data, and it has been shown to outperform linear DR techniques [39, 40].

In Figure 5, we visualise the two dimensional $t$SNE projection of our deep embedding. The learned representation encodes knowledge which is capable of segregating cell populations as well as capturing intraclass phenotypic variance. For instance, FIBs are characterised by the presence of two subpopulations, and representative images drawn from these subclusters are displayed in Figure 6. FIBs exhibit high variance in their morphology since they can appear as either "plump" (active) or elongated (inactive) cells [41]. This distinction is evident in Figure 5, testifying how our deep representation is capable of discovering phenotypic signatures amongst cell subpopulations. A similar phenomenon is observed, to a lesser extent, for the SYN population, and images sampled from these two subclusters are displayed in Figure S6.

## 5   Discussion

We developed and presented a computational pipeline to analyse placental histology imaging. We designed an embarrassingly parallel framework, exploiting deep learning models to solve two independent computer vision tasks. Our nuclei locator can account for the high cellular morphological variance characterising placental tissues, to detect nuclei robustly. We developed a cell classifier capable of accurately (89%) segregating cells within five populations. Furthermore, our approach learned a deep embedding encoding phenotypic knowledge enabling the stratification of cell populations into subclusters of distinct morphological signatures.

There are several avenues for future improvement of our deep learning pipeline. The SYN population is stratified into subclasses since SYNs can appear as either well defined cells, or densely packed SYN aggregates known as syncytial knots (Figure S6). SYN knots are of great biological interest:

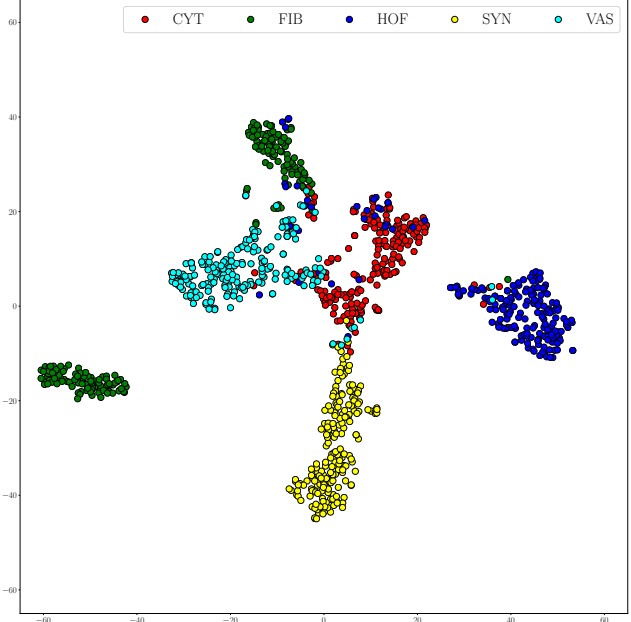

Figure 5: *t*SNE representation of our learned embedding. The *t*SNE projection of test data. A clear separation into five cell types is observed, as well as stratification into subpopulations not explicitly annotated.

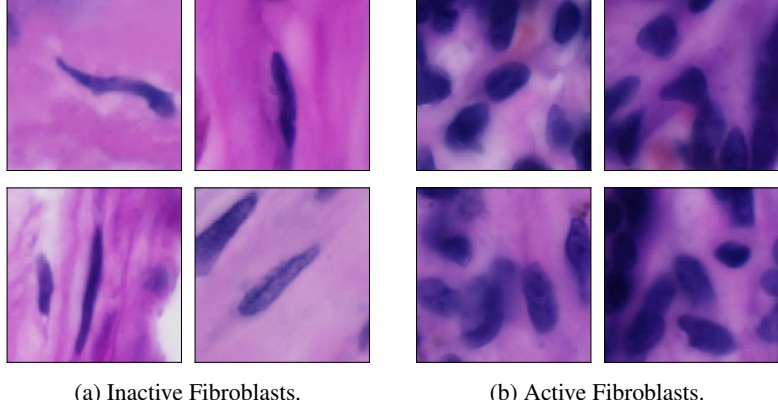

(a) Inactive Fibroblasts.        (b) Active Fibroblasts.

Figure 6: **FIB subpopulations.** In panel (a), we have representative examples drawn from the FIB subcluster on the left of the *t*SNE plot. In (b) we have images from the subpopulation located at the top of Figure 5, the FIB cluster closer to CYT data points.

by analysing their morphology it is possible to discriminate healthy placentas from those affected by anomalous villi maturation [42]. Contrary to individual SYNs, syncytial knots are challenging to annotate due to H&E stain saturation. Therefore, in its current state, our localisation module is unable to capture most nuclei located within knots. Additional work could address this limitation by treating syncytial knots as an additional object for detection with RetinaNet.

Spatial context when annotating histology is important for both trained pathologists and CNNs when conducting inferences about cell type classification. For example, in Figure 4, a monocyte that is present in the vasculature and surrounded by red blood cells is misclassified as VAS. With larger training sets, we expect our model (and any given deep CNN) to become invariant to this, with learned representations being more specific to precise VAS cellular morphology. Finally, whilst RetinaNet is trainable end to end it still has two hyperparameters that need to be tuned, the posterior

probability used for classification and the number of bounding boxes predicted per image. For our application, nuclei number can vary by an order of magnitude across images. Further work could explore learning the total number of objects per image as a regression task at test time.

For our ensemble model, the overall error rate could be reduced by boosting HOF accuracy, currently at 75%. This can be achieved by reducing the confusion between HOF and CYT examples (Figure S4). Hofbauer cells are macrophages and are often found in close proximity to Cytotrophoblasts [43], therefore it would be possible to increase our model's discriminatory power by encoding more spatial context within training images. Reducing HOF classification error therefore requires further careful analyses that we leave to future work.

There is mounting evidence supporting associations between cellular morphology and biological function [44]. Thus, we also plan to examine whether there are correlations between learned representations and cellular morphometric descriptors, *e.g.* perimeter and area. This line of research would help deepen our understanding of placental cell populations, as well as their morphology, and impact on health. Future work will involve the deployment of the pipeline to thousands of placental histology samples, obtained from both healthy and compromised pregnancies. Objective, unbiased measures of cellular variation will help us to understand the molecular basis of fundamental placental biology, population level variability, and its role and impact on fetal and maternal health.

# 6 Code and Data Availability

This paper comes with a dedicated GitHub repository (`https://github.com/Nellaker-group/TowardsDeepPhenotyping`) where we have deposited the scripts, data, and annotations used to develop the pipeline. All pretrained models are available and could be useful for solving, by means of transfer learning, computer vision tasks involving histology data.

The computational pipeline was developed in Python 3 using the machine learning libraries Scikit Learn [45], Keras [46] and TensorFlow [47].

# 7 Acknowledgements

MF is supported through an MRC methodology research grant (MR/M01326X/1). CN is funded through an MRC methodology research fellowship (MR/M014568/1). CML is supported by the Li Ka Shing Foundation, WT-SSI/John Fell funds and by the NIHR Biomedical Research Centre, Oxford, by Widenlife and NIH (5P50HD028138-27). We gratefully acknowledge support from NVIDIA corporation for the donation of GPUs used in this work. We would also like to acknowledge Fizyr, whose GitHub implementation of RetinaNet was of great help during pipeline development. Thanks to Rocio Ruiz Jiménez for her help with preparation, cutting and staining of samples. Finally, we acknowledge assistance from Zeiss with the imaging of the samples using their Axio Scan Z1.

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

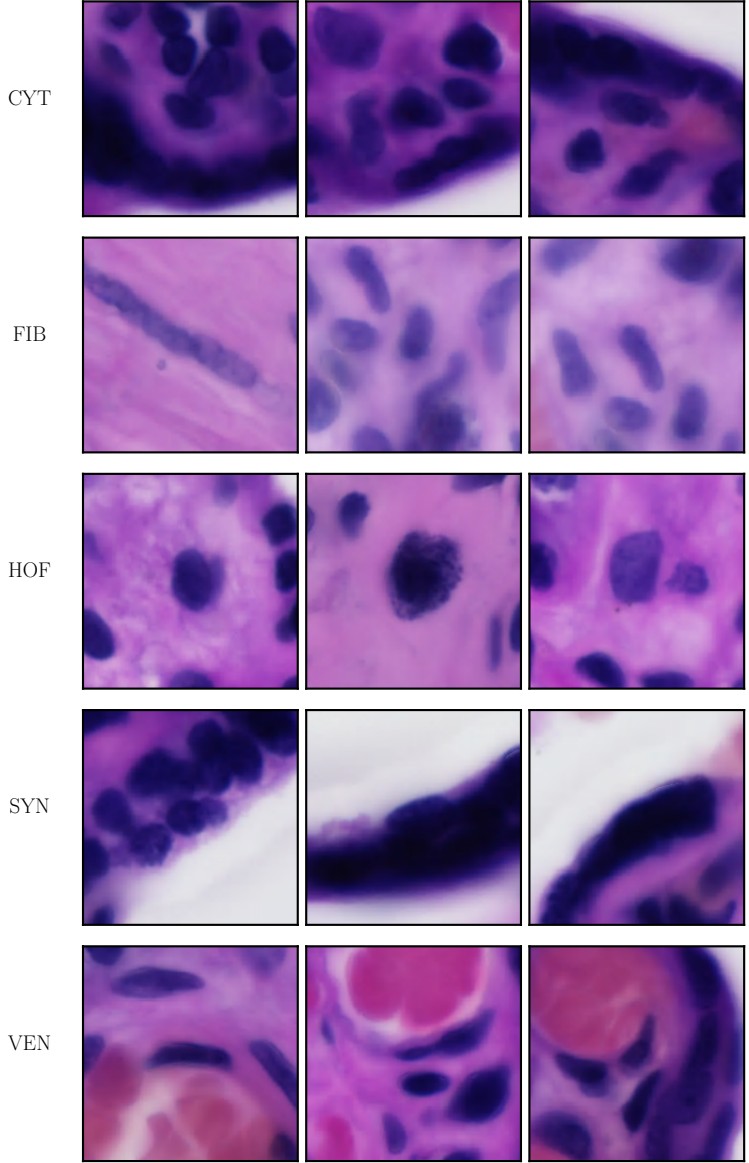

Figure S1: **Representative examples of images used for training the cell classifier.** Each row corresponds to one cell type, displaying multiple examples to capture morphological variance within and across cell populations. Each image is a crop of $200 \times 200$ pixels centred at the location of the nucleus.

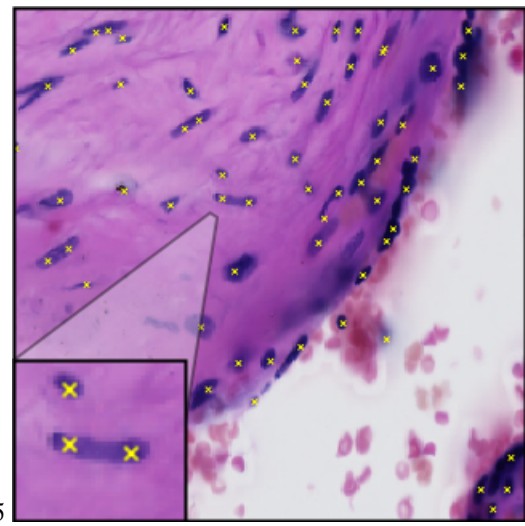

Figure S2: **FCNN-Unet predictions.** Test image showing how FCNN-Unet consistently predicts multiple nuclei for elongated cells (fibroblasts).

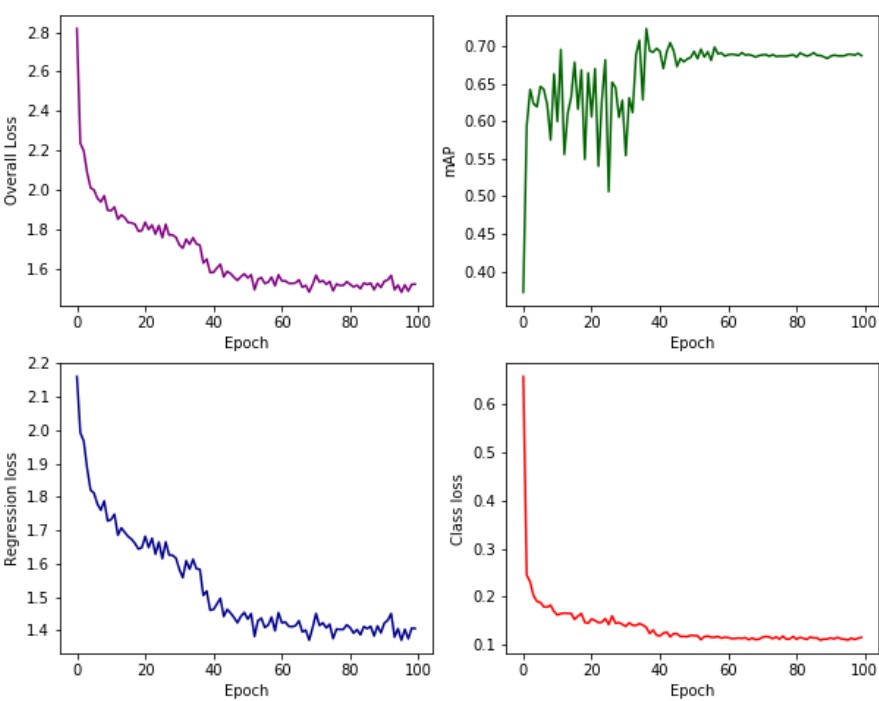

Figure S3: **RetinaNet training losses.** Convergence of all four metrics. Model weights were saved for the best validation mAP.

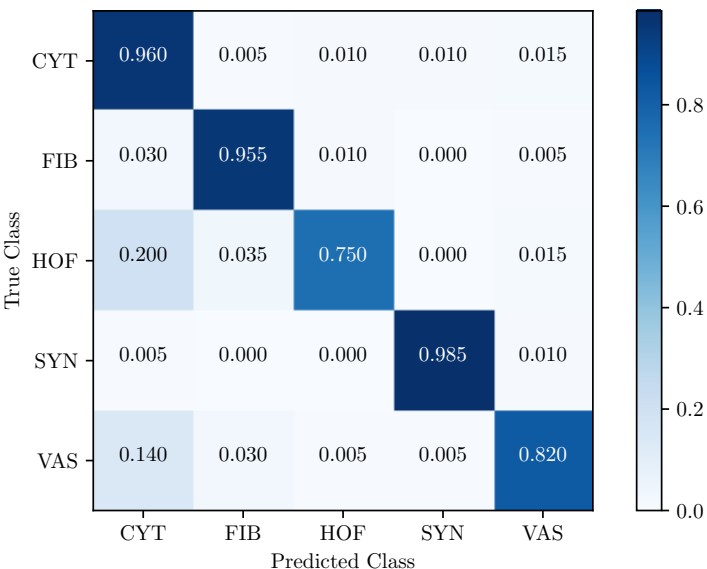

Figure S4: **Ensemble confusion matrix.** By aggregating several base classifiers, we were able to boost test performance across most cell types.

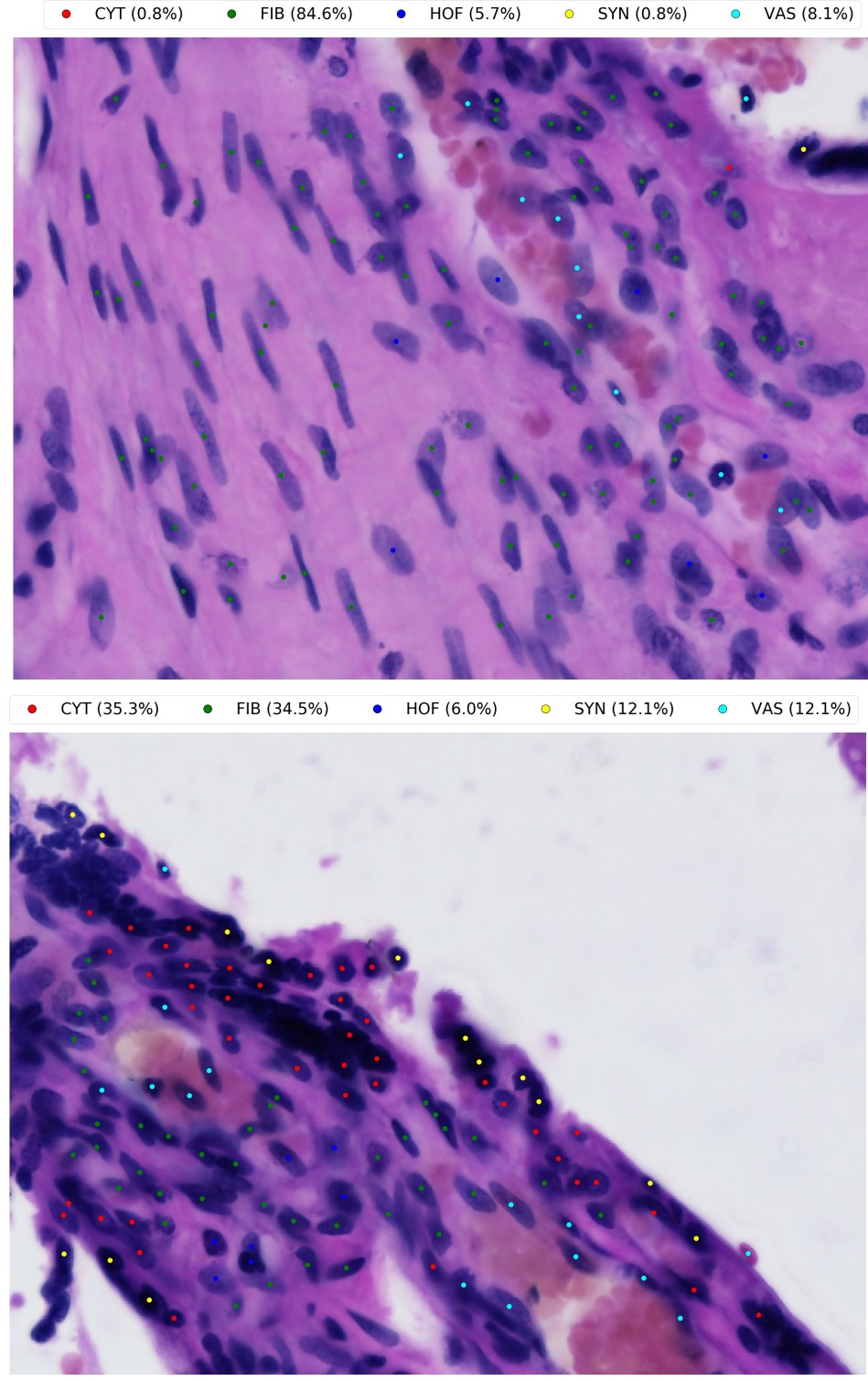

Figure S5: **Test time predictions across whole tiles.** By deploying the pipeline for predicting across whole images, we can learn about local morphology and cellular distributions. For instance, the top panel shows a region deep within the tissue and, consequently, it is enriched with (inactive) fibroblasts.

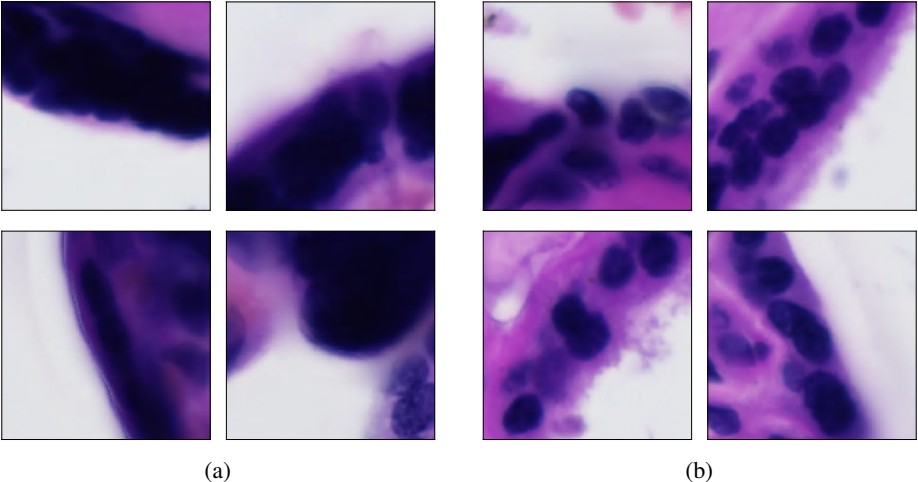

(a)                                  (b)

Figure S6: **Stratification of the SYN population.** Morphological signatures coming from syncytiotrophoblast subpopulations. Panel (a) shows representative images sampled from the SYN subcluster at the bottom of Figure 5, which represent nuclei aggregates known as syncytial knots. Panel (b) displays images drawn from the subcluster closer to the middle of the $t$SNE plot, *i.e.* SYN examples closer to the CYT data cluster.

| Method | Validation Accuracy |
|---|---|
| Bootstrap | 90.4% |
| Class Weights | 90% |
| Down Sampling | 88% |

Table S1: **Performance comparison across data balancing techniques.** We fine tuned InceptionV3 with balanced training sets generated by resampling methods (bootstrap and down sampling), and by penalising missclassification based on class sizes (using class weights).

