# OpenReview forum: "Towards Deep Cellular Phenotyping in Placental Histology"
_MIDL.amsterdam/2018/Conference — MIDL 2018 Poster_

### Review · AnonReviewer1 · 2018-05-09
**Application of DL to an interesting task, but evaluation does not meet the standards**

**Rating:** 2
**Confidence:** 2

**Review:**

The authors propose a deep learning--based approach to the analysis of plactental cells, which consists of a nuclei localization and a nucleus classification step.

The topic is well introduced and motivated. The contributions of this work are not clearly stated. From my point of view however, the contributions are the application and comparison of existing deep learning approaches to a new data set. The data set seems to have a rather small but still reasonable size to train and test the proposed approach. The authors are aware of the limited size and also its imbalance and tackle these issues explicitly using transfer learning and bootstrapping.

For the nuclei localization task, RetinaNet and FCNN-Unet were applied and compared. The comparison is made in textual form, it would have been good to also state quantitative results for the FCNN-Unet. The evaluation of the RetinaNet is comprehensive, but the quality measure mAP should have been explained or at least referenced. This measure is assumed to be precision, thus a measure of recall would have been helpful as well.
The authors also compared two data augmentation methods (with and without stain transformation). They state that performance was improved by applying stain transformation, however, for the relevant cut-offs of the posterior probability, the improvement is not or only very marginally present.
The major criticism here is that Table 1 shows validation quality instead of test quality. Although test quality is given in the text, this should have been the other way round, as validation quality is much less relevant to get insights into the quality of the approach. Additionally, the validation quality is at 0.80 while test quality is at 0.66. This indicates severe over fitting to the validation set, as the optimal model is chosen based on the validation set. However, this issue is not mentioned in the paper and the discussion just states that the approach detects nuclei robustly.

For the nuclei classification task, the InceptionV3 network is applied and produces good results. The authors visualized that in a confusion matrix. Additionally, an ensemble of the InceptionV3, an InceptionResNetV2 and an Xception network is applied to further improve performance. The performance however improved only marginally at the cost of increasing the complexity of the approach.
Additionally, the authors incorporated t-SNE to generate a visualization of the classes in the deep feature space. In this visualization, the classes are well separable. The visualization is very interesting and helpful to corroborate the quality measures of the nuclei classification. With the visualization, the authors were able to discovered that the FIB class in fact consists of two subclasses and provide a medical explanation.

Summary of pros(+) and cons(-):
+ limits of data set noticed and tackled
+ Use of t-SNE visualization interesting and validates learned features
- Quality measures are not described
- Validation quality is highlighted instead of test quality
- Clear over fitting not explained or addressed in any way

**Special Issue:**

No

---

> ### Comment · ~Michael_Ferlaino1 · 2018-06-29
> **Response to Reviewer #1**
>
> We thank the reviewer for their comments. Please find below our responses, along with the corresponding modifications implemented in the manuscript.
>
> C1. For the nuclei localization task, RetinaNet and FCNN-Unet were applied and compared. The comparison is made in textual form, it would have been good to also state quantitative results for the FCNN-Unet. The evaluation of the RetinaNet is comprehensive, but the quality measure mAP should have been explained or at least referenced. This measure is assumed to be precision, thus a measure of recall would have been helpful as well. The authors also compared two data augmentation methods (with and without stain transformation). They state that performance was improved by applying stain transformation, however, for the relevant cut-offs of the posterior probability, the improvement is not or only very marginally present. The major criticism here is that Table 1 shows validation quality instead of test quality. Although test quality is given in the text, this should have been the other way round, as validation quality is much less relevant to get insights into the quality of the approach. Additionally, the validation quality is at 0.80 while test quality is at 0.66. This indicates severe over fitting to the validation set, as the optimal model is chosen based on the validation set. However, this issue is not mentioned in the paper and the discussion just states that the approach detects nuclei robustly.
>
> R1. RetinaNet outputs bounding boxes and posteriors quantifying the confidence the model has in the box actually containing a nucleus. Such posterior probability is an additional hyperparameter that needs to be tuned before deploying the model on the test set. In table 1 we show how the validation accuracy varies across different posterior thresholds. Only once the best value of the hyperparameter is found on validation data, it is possible to deploy the model on the test set in order to assess generalisation performance. We believe this is what we describe in Section 4.1, and we report only one test accuracy as we are only estimating the performance of one model. Additionally, we believe the gap between validation and test accuracies is partly driven by our data splitting protocol. Test nuclei were annotated only from one individual (one WSI out of ten), not used to annotate training and validation examples. With this protocol we were trying to replicate what can happen at test time, where RetinaNet could be deployed to localise nuclei in images generated with an experimental protocol different from ours. This is also one of the reasons behind our development of stain transformations, as we wanted our model to be robust to stain variance across images generated using different experimental protocols.  This could be seen as an instance of domain adaptation, and we tried to implement new data augmentation transformations to improve generalisation to unseen images, beyond the ones available in our pilot project. Furthermore, we have extensively expanded Section 4.1 in order to clarify the issues raised.

---

### Review · AnonReviewer3 · 2018-05-12
**Solid work on cell detection and classification**

**Rating:** 4
**Confidence:** 1

**Review:**

This paper describes a solid piece of work for detecting and classifying cells of 5 different cell types in placenta H&E slides. The system contains two parts: detecting nuclei as a cell candidate detector, and classifying the detected cells in one of 5 types.

The data set was collected by the authors. 10 slides were scanned and tiles of 1600x1200 were cropped. 13k nuclei were annotated, 9.k cells were classified. The data set is not shared as far as I can see, although the github repo with the code, that is shared, contains some images. The authors could clarify if they intend to share the data. I think it is likely results can be substantially improved, and sharing the data could allow the community to assist in this effort.

For the detection step two published approaches have been attempted, Retinanet, ref 22, and FCNN, ref 25. The authors choose Retinanet because FCNN has the issue that often two nuclei are detected in the nucleus is elongated. This is shown in Figure S2. I think it should be fairly easy to merge such double detections into one.

The results of the nuclei detection phase, with Retinanet, are not clearly described. Table 1 is not informative, or at least not easy to interpret. The text mentions a generalization performance of 0.66, but this is not an entry in the Table. The standard for evaluating detection tasks is to show FROC curves. My impression is that the performance is not very good. Figure 2 contains 9 cells that are missed. It seems the system consistently fails on detecting smaller nuclei. There is also one example where a detection bounding box covers two smaller reference boxes. How is the hit criterion defined, i.e., how do you handle this? Papers on detection tasks in medical image analysis tend to describe this in detail. There are no examples of false positive detections, surely the system produces these occasionally too. So Figure 2 alone is not a very good way of illustrating results.

Considering all the missed detections of Retinanet, I wonder if FCNN with a postprocessing step might not be better, or al least have additional value if the two approaches were ensembled.

The description of the classification step results suffers from similar issues. Why show a confusion matrix in the main text of only Inception V3, followed by a table of the ensembling experiments, and have another confusion matrix in the supplementary figures? You could easily have the two matrices side by side. You write "Misclassification error types are comprehensively visualized in Figure S4." but this figure is a confusion matrix and is not visualizing anything. It would be very interesting to see images of misclassified cells. 88% or 89% accuracy indicates there may be substantial space for improvements.

It would be great to compare the results with independent human expert classifications.

The subsequent results are preliminary, the discussion lists sensible next steps. To summarize the work as "a computational pipeline to comprehensively analyze placental histology imaging" is an overstatement, you have made a promising start in detecting and classifying cells, a comprehensive analysis of placental histology would be the next step.

Pro:
* Solid work
* Very well written; a joy to read
* Probably a first attempt to analyze placental H&E slides in this manner; novel and relevant application

Con:
* The title is misleading. Deep Cellular Phenotyping sounds fantastic, but that is not covering the content of the paper. In general, the paper is overselling.
* Results of the detection seem rather poor and are not compared with human performance
* Result of the classification are not critically analyzed and not compared with human performance

**Special Issue:**

No

---

> ### Comment · ~Michael_Ferlaino1 · 2018-06-29
> **Response to Reviewer #3**
>
> We are grateful to the reviewer for their suggestions, and we are really pleased to know they found the manuscript a joy to read. Please find below our responses, along with the corresponding amendments to the manuscript.
>
> C1. [...]. The data set is not shared as far as I can see, although the github repo with the code, that is shared, contains some images. The authors could clarify if they intend to share the data. I think it is likely results can be substantially improved, and sharing the data could allow the community to assist in this effort.
>
> R1. All data are freely available on the GitHub repository accompanying the paper. These include the images to train, validate and test, the nuclei annotations, cell labels and patches, and model weights. All of  these data can be found at https://github.com/Nellaker-group/TowardsDeepPhenotyping.
>
> C2. The results of the nuclei detection phase, with Retinanet, are not clearly described. Table 1 is not informative, or at least not easy to interpret. The text mentions a generalization performance of 0.66, but this is not an entry in the Table. The standard for evaluating detection tasks is to show FROC curves. My impression is that the performance is not very good. [...].
>
> R2. RetinaNet outputs bounding boxes and posteriors quantifying the confidence the model has in the box actually containing a nucleus. Such posterior probability is an additional hyperparameter that needs to be tuned before deploying the model on the test set. In table 1 we show how the validation accuracy varies across different posterior thresholds. Only once the best value of the hyperparameter is found on validation data, it is possible to deploy the model on the test set in order to assess generalisation performance. We believe this is what we describe in Section 4.1, and we report only one test accuracy as we are only estimating the performance of one model. Furthermore, we have extensively expanded the section where we discuss RetinaNet's performance in order to clarify the points raised in the comment. Regarding FROC curves, it is standard to assess bounding box regression tasks by means of mean average precision (mAP), and this is the metric we used in the present work.
>
> C3. Considering all the missed detections of Retinanet, I wonder if FCNN with a postprocessing step might not be better, or at least have additional value if the two approaches were ensembled.
>
> R3. We agree with the reviewer as it would be extremely interesting to investigate further whether it’s possible to boost the performance of fully convolutional neural networks for nuclei detection. This line of investigation was not included as this line of enquiry wasn’t the main objective of the pilot project.
>
> C4. The description of the classification step results suffers from similar issues. Why show a confusion matrix in the main text of only Inception V3, followed by a table of the ensembling experiments, and have another confusion matrix in the supplementary figures? You could easily have the two matrices side by side. You write "Misclassification error types are comprehensively visualized in Figure S4." but this figure is a confusion matrix and is not visualizing anything. It would be very interesting to see images of misclassified cells. 88% or 89% accuracy indicates there may be substantial space for improvements.
>
> R4. We agree that the visualisation of misclassified examples could provide insight into the types of error made by the model, thereby allowing its improvement, and we aim to include this in an updated version of the manuscript. This is another line of enquiry we are currently pursuing, after delivering the results obtained during the pilot project in the present publication. Furthermore, within section 4.2, we collected the confusion matrix as well as classification report for InceptionV3 with the aim to comprehensively visualise our model performance, beyond just using overall and class accuracies.
>
> C5. It would be great to compare the results with independent human expert classifications.
>
> R5. We agree with the reviewer as it would be interesting to compare the accuracy of our model with human experts. However, we felt this line of investigation was outside the scope of the current work.
>
> C6. The subsequent results are preliminary, the discussion lists sensible next steps. To summarize the work as "a computational pipeline to comprehensively analyze placental histology imaging" is an overstatement, you have made a promising start in detecting and classifying cells, a comprehensive analysis of placental histology would be the next step.
>
> R6. We agree with this comment and, also in line with recommendations from Reviewer #4, we have removed "comprehensively" from the current version of the manuscript.

---

### Review · AnonReviewer4 · 2018-05-17

**Rating:** 4
**Confidence:** 3

**Review:**

The authors suggest models for localizing (RetinaNet vs. FCNN) and classifying (Inception V3 + Inception ResNetV2 + Inception) placental cells. Care was taken on evaluating preprocessing steps (stain transformation), boosting performance using ensembles, and on balancing the data using bootstrap, class reweighting, and downsampling. The embedding of the learned classification representations is analyzed with tSNE and subclusters are visualized to gain insights into the subpopulations of the data.

I enjoyed reading this paper and am satisfied with the clarity and structure of the manuscript. The reported results look encouraging and the choice of model and parameters seem reasonable. I have a couple of remarks:

- I appreciate that the code will be publicly available.

- It seems that the test images were extracted from around 8 individuals (approx. 1119/(13179/91)). Is this correct? This would be quite a small number of test individuals. If true, I would suggest to perform cross validation for improving the evaluation.

- How was the confidence of the annotators assessed (Section 3.1.2)?

- How were the hyperparameters (e.g., perplexity) for the tSNE visualization selected (Figure 5)?

- I would recommend to entirely leave out the word "comprehensively" and leave this assessment up to the reader.

**Special Issue:**

No

---

> ### Comment · ~Michael_Ferlaino1 · 2018-06-29
> **Response to Reviewer #4**
>
> We thank the reviewer for their helpful comments, and we are really pleased to hear they enjoyed the reading of our manuscript. Please find below our responses to the comments, along with the corresponding modifications added to the latest version of the manuscript. In addition, the code, images and all annotations have been and are publicly available since our initial submission at https://github.com/Nellaker-group/TowardsDeepPhenotyping
>
> C1. It seems that the test images were extracted from around 8 individuals (approx. 1119/(13179/91)). Is this correct? This would be quite a small number of test individuals. If true, I would suggest to perform cross validation for improving the evaluation.
>
> R1. For this pilot project, we collected 10 placenta samples from 10 different women, who gave birth at term to healthy babies. This consists of millions of images as each histology slide is in the range of 100,000 X 200,000 pixels. We are currently planning the deployment of the pipeline to population studies, where we will collect thousands of placenta samples from both healthy and compromised pregnancies. For the nuclei detection task, train, validation, and test data were annotated by sampling a different individual for each set. This means that test nuclei were annotated from one single individual, and different from the individuals sampled to generate the training and validation folds. We implemented such a protocol with the aim of preventing information leakage from the training set into the test fold. For example, by training and testing the model using tiles from the same individual, the nuclei detector performance could be boosted by using information about stain gradient and concentration. Therefore, our training protocol would have required the implementation of cross validation, by leaving one different individual out to generate the test set at each loop step. We decided not to implement such an approach due to the high computational burden.
>
> C2. How was the confidence of the annotators assessed (Section 3.1.2)?
>
> R2. We did not set up a protocol to quantitatively assess the confidence of annotators. However, in order to control the false positive/negative rates, we required clinicians to only annotate cell examples they could label with high confidence. We believe this procedure enabled the annotation of a relatively small, but very high quality data set.
>
> C3. How were the hyperparameters (e.g., perplexity) for the tSNE visualization selected (Figure 5)?
>
> R3. We obtained 2D visualisations of deep embeddings by using the tSNE implementation in Scikit Learn with the default, recommended values for the hyperparameters.
>
> C4. I would recommend to entirely leave out the word "comprehensively" and leave this assessment up to the reader.
>
> R4. We thank the reviewer for their suggestion, and we have now removed the word "comprehensively" from the manuscript.

---

### Decision · Program_Chairs · 2018-05-15
**Paper10 Acceptance Decision**

Poster